# Research Strategies to Develop Environmentally Friendly Marine Antifouling Coatings

**DOI:** 10.3390/md18070371

**Published:** 2020-07-18

**Authors:** Yunqing Gu, Lingzhi Yu, Jiegang Mou, Denghao Wu, Maosen Xu, Peijian Zhou, Yun Ren

**Affiliations:** 1College of Metrology &Measurement Engineering, China Jiliang University, Hangzhou 310018, China; guyunqing@cjlu.edu.cn (Y.G.); s1902080449@cjlu.edu.cn (L.Y.); wdh@cjlu.edu.cn (D.W.); msxu@zjut.edu.cn (M.X.); 19a0205122@cjlu.edu.cn (P.Z.); 2Zhijiang College, Zhejiang University of Technology, Shaoxing 312030, China; renyun_ry@hotmail.com

**Keywords:** antifouling mechanism, antifouling coating, antifoulant, environmentally friendly, polymer

## Abstract

There are a large number of fouling organisms in the ocean, which easily attach to the surface of ships, oil platforms and breeding facilities, corrode the surface of equipment, accelerate the aging of equipment, affect the stability and safety of marine facilities and cause serious economic losses. Antifouling coating is an effective method to prevent marine biological fouling. Traditional organic tin and copper oxide coatings are toxic and will contaminate seawater and destroy marine ecology and have been banned or restricted. Environmentally friendly antifouling coatings have become a research hotspot. Among them, the use of natural biological products with antifouling activity as antifouling agents is an important research direction. In addition, some fouling release coatings without antifoulants, biomimetic coatings, photocatalytic coatings and other novel antifouling coatings have also developed rapidly. On the basis of revealing the mechanism of marine biofouling, this paper reviews the latest research strategies to develop environmentally friendly marine antifouling coatings. The composition, antifouling characteristics, antifouling mechanism and effects of various coatings were analyzed emphatically. Finally, the development prospects and future development directions of marine antifouling coatings are forecasted.

## 1. Introduction

Marine fouling organisms refer to the general term for all kinds of marine organisms attached to the surface of marine facilities and causing damage to human marine economy [1]. Marine fouling organisms not only endanger the development of the marine economy, but also hinder human exploitation of the ocean. At present, more than 4000 marine fouling organisms have been discovered [2], such as microorganisms mainly including bacteria, diatoms and Ulva spores, etc.; large fouling organisms mainly include barnacles, bryozoans, mussels and algae.

Various fouling organisms easily attach to ships, oil platforms and breeding facilities in the ocean [3]. Barnacles and other invertebrates are firmly attached to the hull surface by secreting a biological adhesive. As the barnacle grows, its edges will destroy the anticorrosion layer on the hull surface, thereby accelerating the hull corrosion [4]. As shown in Figure 1 [5], when the hull surface is attached, fouling organisms will corrode the hull surface and increase the roughness of the hull surface, thereby reducing the speed of the ship [6,7,8], increasing fuel consumption and greenhouse gas emissions [9,10]. In recent years, with the rapid development of offshore industries such as submarine oil and marine power generation, the damage of marine fouling organisms to man-made facilities has become more severe. For example, when an offshore oil platform is attached, it will increase the weight of the facility and weaken its ability to resist tsunami and storm risks. In addition, fouling organisms can block drainage pipes, jeopardizing the safety and service life of marine facilities. Some marine fouling organisms enter the non-native land by attaching to the bottom of the ship, thereby causing biological invasion and causing great harm to the global marine ecology [11]. Every year, a lot of money is invested in ship surface cleaning and maintenance of marine facilities. With the continuous development of the human marine economy, the economic losses caused by marine fouling organisms are getting larger and larger, thus, effective and economic prevention methods are getting more and more attention.

The key to managing biological attachment is to block biological attachment from the source. Applying antifouling coating is currently the easiest and most widely used antifouling method. Traditional antifouling coating often use poisonous drugs to poison fouling organisms, the main components of which are organic tin [12] and cuprous oxide. However, the accumulation of metal elements in metal compounds in fish and shellfish will cause biological variation and death and pollute seawater, and thus harm the marine ecological environment. Taking into account the greater harm of organic tin to the marine environment [13], organic tin coatings have been banned worldwide. Although cuprous oxide is less toxic, as it continues to accumulate, its impact on marine ecology will become greater and greater. Therefore, the development of efficient and environmentally friendly marine antifouling coatings has become a research hotspot [14].

There are two main research directions for new antifouling coatings. One direction is to develop non-toxic and environmentally friendly antifoulants that replace metal compounds, that is, to find antifouling active substances from the ocean. An antifoulant developed by marine natural products can effectively reduce the harm to the environment, which is also one of the hot issues in current research [2,15]. In addition, some terrestrial biological products have also attracted attention. The purification or modification of natural products through chemical methods can obtain antifoulants with extremely strong antifouling capabilities. The second direction focuses on fouling release coatings, which will modify the surface characteristics of the material, making it difficult for fouling organisms to adhere or make them easier to remove after attachment. Due to the complex diversity of the marine environment, it is becoming more and more difficult for a single mechanism of antifouling coating to meet the requirements of use, and some composite antifouling coatings that combine multiple antifouling principles have also appeared. With the development and application of nanotechnology and polymer materials, the antifouling ability of antifouling coatings has been further improved [16].

This article first reveals the fouling mechanism of marine fouling organisms. On this basis, it mainly summarizes the development status of environmentally friendly antifouling coatings in three aspects from antifoulant type antifouling coatings, fouling release antifouling coatings and other important antifouling coatings. Our analysis reveals the antifouling mechanism of various antifouling coatings, and reviews the advantages and disadvantages of the new coating. Finally, the development trend of marine antifouling coatings is forecasted.

## 2. Formation Mechanism of Fouling

In order to better solve the problem of marine biological fouling, it is necessary to understand how the attachment of marine fouling organisms to the surface of objects occurs. Marine biological fouling is a process where biological communities accumulate on the surface of materials. The process is complex. As shown in Figure 2, biological fouling can be divided into three stages: conditioned film, micro-fouling and macro-fouling.
(1)Conditioned film [17]: When a clean surface is placed in seawater, a layer of organic matter, including polysaccharides, lipids and protein molecules, will accumulate within minutes. The physical adsorption of these molecules on the surface of the material causes the accumulation of organic molecules to form a thin film, which is usually called the conditioned film. This adsorption is reversible, relying only on weak non-covalent bond forces, such as van der Waals forces, electrostatic forces and hydrogen bonding.(2)Micro-fouling [18,19]: After the conditioned film is formed, bacteria and diatoms and other microorganisms will adhere to the conditioned film within 24 h, and floating bacteria will gather on the surface. These attached bacteria and algae will secrete new extracellular polymers (EPS) in order to further improve their fixation ability with the material surface or conditioned film, thereby forming a biofilm composed of water, organic matter, microorganisms and extracellular metabolites. This process is irreversible, and this process is called micro-fouling.(3)Macro-fouling [20]: The formation of micro-fouling can further aggravate the generation of biological fouling. Biofilms provide abundant food and nutrition for the attachment of larvae of multicellular species and large marine organisms. Barnacles, shellfish and other large fouling organisms adhere to and grow within a few days, and complex biomes slowly form, and large-scale fouling will form in a few months. This is called macro-fouling. This process can be completed within 1–2 months and organisms can be attached to the surface for several years.

However, this model cannot apply to all marine fouling organisms. The actual fouling of the ocean is sometimes not in this order. For example, the cyprids of barnacle *Amphibalanus amphitrite* can adhere to the surface of materials without biofilm.

## 3. Marine Antifouling Coating

Thus far, the application of antifouling coating is the most effective means of antifouling. Self-polishing copolymer-based coatings containing TBT had high efficiency in preventing the settlement and growth of marine fouling organisms, thus they were once used widely. However, they were globally banned in 2008 due to their persistent toxicity to non-target organisms. It is time to develop environmentally friendly systems to prevent marine biofouling. Table 1 summarizes the current popular research directions of environmentally friendly marine antifouling coating.

### 3.1. Natural Product Antifoulant

Cuprous oxide antifouling coatings have been used widely in today’s marine ship. However, it also has potential environmental risks. Because the copper element in the cuprous oxide coating is enriched in the ocean, it causes a lot of death of seaweed and destroys the ecological balance, and will eventually be restricted or banned. In order to prevent the antifouling coating from harming the ecological environment, the researchers tried to find useful substances from nature to replace copper-based coatings. At present, the research of new antifoulants mainly focuses on the extraction and optimization of natural products. Many natural substances extracted from animals and plants have a good antifouling effect [21]. Compounds such as terpenoids, steroids, carotenoids, phenolics, furanones, alkaloids, peptides and lactones extracted from the ocean all have antifouling activity [22,23]. Irritation substances extracted from terrestrial plants such as oleander and pepper are also important sources of antifoulants.

#### 3.1.1. Marine Products with Antifouling Activity

Some marine organisms produce certain metabolites that inhibit fouling biological activity, which helps to develop pollution-free antifoulants composed of natural products [24]. In the past few years, some marine biological products with anti-pollution function have been discovered [25,26]. For example, Zhang et al. [27] discovered subergorgic acid (SA) from *Subergorgia suberosa*, which proves that it is non-toxic and has a strong inhibitory effect on attachments. In addition, Zhang et al. [28] conducted an in-depth study on the structure-activity relationship of SA and found that both ketone carbonyl and the double bond are necessary elements with antifouling activity. The experimental scheme is shown in Figure 3. Firstly, the esterification product 1 was readily obtained from SA using CH_3_I and K_2_CO_3_; secondly, NaBH_4_ or LiAIH_4_ was used to reduce ketone carbonyl to hydroxyl group; thirdly, LiOH(3eq)/TFH/H_2_O was used, trying to remove the methyl to give 3; finally, a strong oxidant 30%H_2_O_2_/TFA was used to modify the double bond of SA into two hydroxyl groups to obtain product 4. Products 1 and 2 still have antifouling activity, while products 3 and 4 have no antifouling activity, thus, ketone carbonyl and double bond groups are necessary elements for antifouling effect. On this basis, benzyl esters and methylene chains with proven antifouling activity were added to SA to form new SA derivatives. According to the methods depicted in Figure 3b, with dry DMF as solvent and K_2_CO_3_ as base, SA can smoothly react with various benzyl halides and quantitatively obtain the desired benzyl ester. As shown in Figure 3c, using SA and dibromoalkanes of various lengths as starting materials, compounds 15–20 were easily synthesized in good yields. The antifouling test results show that the antifouling effect of all benzyl esters of SA is basically the same or stronger than that of SA containing unsubstituted benzyl rings. The results also show that the antifouling effect of SA derivatives containing methylene chains is not as good as that of SA, and the influence of the length of methylene chains on the antifouling effect of SA derivatives depends on the functional group at the opposite position of the methylene chain. The impact of this type of antifouling agent on non-target marine life needs further testing. However, the antifouling active functional groups proved in its experiments have certain value for the research of antifouling compounds.

In nature, marine organisms have evolved many biological strategies to interact with microorganisms to protect themselves from pathogens or from being parasitized. [29]. In particular, sponges and their associated microbiota produce compounds that interfere with quorum sensing (QS) mechanisms [30,31]. Quorum sensing is a synchronization mechanism within a bacterial population. The bacteria use QS to communicate, regulate their behavior and assess their population density. The use of marine biological secretions to interfere with bacterial QS can effectively inhibit the formation of bacterial biofilms, making it difficult for other organisms to attach [32]. Tintillier et al. [33] studied the marine sponge *Pseudoceratina sp. 2081* and isolated four new tetrabromotyrosine derivatives exhibiting antifouling and quorum sensing inhibition (QSi) properties. Structures of the isolated bromotyrosine metabolites is showed in Figure 4. They tested the antifouling activity of 6 kinds of extracts. Among the six compounds tested, the most active were 1, 3 and 5. The mode of action of those compounds are not based on toxicity. They operate through a targeted mode of action on bacteria and microalgae, as they only affect adherence and not growth.

Indole derivatives found in bryozoans and ascidians have been shown to have antifouling effects [34]. However, due to the uncontrollable release rate of indole derivatives in antifouling coatings, the effect of antifouling coatings with added indole derivatives is poor. Feng et al. [35] prepared acrylate resins suspending the indole derivative structure in their side chain by the free-radical polymerization. Its preparation method and antifouling principle are shown in Figure 5. The prepared indole derivatives have good antibacterial and algae inhibiting effects. The self-polishing rate of the acrylate resin polymerized with indole derivatives is reduced, thereby obtaining a long-term self-polishing effect. The dynamic simulation measurements and static immersion measurements of the coating were performed to study in the marine environment. The results show that acrylate resin containing indole derivatives has long-lasting efficacy in biological control and the acrylate resins suspending the indole derivative structure in their side chain can make fouling organisms difficult to grow. In addition, indole derivatives inhibit the marine algae growth by interfering with the equilibrium of calcium ions in algal cells and decreasing the abundance of cellular Ca^2+^ [36]. Under the action of seawater, the acrylic resin continuously renews its surface through the hydrolysis of ester groups. This self-polishing property makes attached organisms easily fall off the surface, exposing new indole derivative structures. Therefore, the combination of these two functions of the resins can significantly improve their antifouling properties. Since indole derivatives are easily degraded by microorganisms, they will not cause a great impact on the ecological environment.

Most marine fouling organisms secrete bio-gum to enhance surface adhesion. Bio-glue is a protein, and many biological enzymes in nature can degrade proteins. The essence of biological enzyme is protein, which is easy to decompose, causes no pollution and will not damage the marine ecology, thus, the biological enzyme with antifouling function is an ideal antifoulant. Wang et al. [37] isolated a marine proteolytic bacterial strain of *Bacillus velezensis* from sea mud and found that the protease produced by it had obvious inhibitory effects on barnacles, diatoms and mussels. This method uses bacteria to obtain proteases, which has the possibility of mass production. However, because marine fouling organisms secrete complex binders, polysaccharides and even lipids, it is difficult to control marine fouling by a single protease. Therefore, in the application research of enzyme-based environmental protection antifouling materials, it is necessary to use a combination of several enzymes to control marine antifouling.

Currently, these marine biological antifoulant usually cannot obtain a sufficient amount from the ocean, and are difficult to chemically synthesize at low cost, thus, to a certain extent, it is still difficult to commercially mass-produce. However, it is important to develop a more perfect antifoulant by studying its antifouling mechanism through chemical methods and exploring the functions of various groups.

#### 3.1.2. Terrestrial Products with Antifouling Activity

Terrestrial organisms are also a valuable source of natural antifoulants. Compared with marine life, many terrestrial plants are easy to produce commercially because of their wide distribution or large-scale cultivation [38]. Liu et al. [39] isolated four cardenolides from *Nerium oleander L*. and evaluated their antifouling activity and toxicity to non-target organisms. The result showed that all of the tested compounds showed a strong inhibitory activity against barnacle settlement, with very low or moderate toxicity to non-target organisms. Their antifouling effect far exceeds that of TBT, Irgarol and copper. However, before it is widely used, its degradability and toxicity to non-target organisms need further study.

Chalcone is a compound commonly found in natural products. Celery was once the main source of chalcone, and now it is mainly synthesized by chemical means. Almeida et al. [40] studied the synthesis methods of 16 kinds of chalcone derivatives and studied the antifouling properties and ecotoxicity of chalcone through biological experiments. The results show that chalcone can effectively prevent the settlement of mussel larvae and inhibit the accumulation of other fouling microorganisms. In addition, they proved that these compounds have low ecotoxicity and clarified their great potential in the field of marine antifouling. Sathicq et al. [41] evaluated the synthetic furan-based chalcone antifouling coatings. They used furan rings instead of B rings to make a variety of different furylchalcones antifouling coatings, and conducted field experiments. Experimental results show that: using furylchalcones as an antifouling coating is an effective means of preventing marine fouling organisms; the synthesis of furylchalcone is simple and efficient, which is beneficial to reduce economic costs; furylchalcone is degradable, having little impact on the marine environment. Despite this, the mechanism of action of chalcone derivatives as antifoulant is not yet clear, thus, it is necessary to thoroughly understand the toxicity of furanyl chalcone in non-target marine organisms before applying it to the marine environment.

Camphor extracted from the trunk of camphor tree is a terpene natural organic compound, which can also be synthesized in large quantities by chemical means [42]. It is often used to repel insects and mosquitoes in daily life. In theory, it is a potential compound with antifouling capabilities [43]. Borneol extracted from herbs such as lavender and chamomile is a crystalline cyclic alcohol. Borneol-based polymer, isobornyl methacrylate (IBOMA) has been proven to have a strong activity against bacterial infections and can be used to prepare antibacterial coatings [44]. Hu et al. [45] synthesized an environmentally friendly antifouling coating based on antibacterial polymer (IBOMA) and natural antifouling agent (camphor). The prepared coating can slowly degrade in sea water and release borneol. Borneol itself has antibacterial properties [46], and at the same time, it provides a self-renewing surface to prevent dirt from sticking. In addition, camphor was released, providing a special antibiosis and antifouling surface with sterilized polymers to play a synergistic antifouling effect. Due to the hydrolysis of acrylic silicone resin and the increase in carboxylate content after immersion, the hydrophobicity of the coating surface quickly becomes dehydrated. The experiment proves that the produced coating can be controlled and slowly released, and has a continuous effect. Marine testing has shown the great potential of the coating, but the coating still needs a longer time to evaluate. 

Capsaicin extracted from chili is a spicy vanillin amide alkaloid with antibacterial effects and is also regarded as a natural non-toxic antifoulant [47]. Wang et al. [48] conducted extensive research on capsaicin and its derivatives and reported six capsaicin derivatives for the first time, proving that capsaicin and its derivatives have excellent antifouling properties. Liu et al. [49] used capsaicin as an antifoulant and used high-density polyethylene (HDPE) as a substrate to make an antifouling coating by flame spray. Capsaicin, HDPE and coating preparation methods are shown in Figure 6. Compared with the raw materials, the morphology and grain size of capsaicin in the coating have changed significantly, and the coating has significant antibacterial and antifouling capabilities. In addition, capsaicin can be extracted from capsicum relatively easily, thus, this antifouling technology has low cost and high feasibility, and is suitable for widespread promotion.

Biological antifoulants come from a wide range of sources, and the effect is quite excellent. However, the method of extracting it from organisms is still difficult to use on a large scale. Even though some biologically active substances can be artificially synthesized, it is far less convenient than metal compounds. In addition, natural product antifouling agents may have ecotoxicity, that is, they might be toxic to non-target organisms. Therefore, for the research of natural antifouling agents, we should not only pay attention to their antifouling properties, but also explore their ecotoxicity.

### 3.2. Fouling Release Coatings

The fouling release coatings (FRC) itself does not contain antifoulants, mainly to use the dual characteristics of fouling organisms that are difficult to attach to the surface of low surface energy materials or easy to desorb on the surface to prevent or reduce the attachment of marine fouling organism. Compared with traditional self-polishing antifouling coatings (SPC), FRC uses its own physical properties to hinder the attachment of fouling organisms, and will not release antifouling agents into the ocean, thereby not causing pollution to the marine environment.

Biological attachment behavior has a direct relationship with the surface characteristics of materials. Different surfaces have different or even opposite effects on biological adhesion. Low surface energy coatings mainly inhibit the attachment of fouling organisms by physical means, that is, the use of the characteristics that fouling organisms are difficult to attach to the surface of low surface energy materials to prevent or reduce the attachment of marine fouling organisms. Research shows that when the surface energy is less than 25 mJ/m^2^, that is, the contact angle between the coating and the liquid is greater than 98°, and it can effectively reduce the adhesion of fouling organism [50]. At present, this type of antifouling coating mainly includes two major categories of silicone and organic fluorine.

#### 3.2.1. Organic Fluorine

In fluorine-containing polymers, due to the strong electronegativity of fluorine atoms, low polarizability and high C-F bond energy (460 kJ/mol), these materials are given high chemical stability and oleophobic and hydrophobic properties [51]. Arukalam et al. [52] prepared a perfluorodecyltrichlorosilane-based poly (dimethylsiloxane)-ZnO (FDTS-based PDMS-ZnO) nanocomposite coatings with surface energies within 20–30 mN/m for possible antifouling and anti-corrosion applications. ZnO is that the coating has antibacterial ability and changes the roughness of the coating surface, thereby improving its anti-adhesion ability. FDTS can modify the surface energy of the coating to keep its surface energy within 20–30 mN/m. Electrochemical impedance spectroscopy (EIS) test shows that the coating has excellent corrosion resistance. These characteristics mean that this coating has a good application prospect in the field of marine antifouling. Xu et al. [53] prepared a polymer with pendant branched poly (ethylene glycol) (PEO) and poly (2,2,2-trifluoroethyl methacrylate) (PFMA) structural units. The structure is shown in Figure 7. The b-PFMA-PEO asymmetric molecular brush with its side chains densely distributed with the same repeat unit is used to prepare a fluorine-containing synergistic nonfouling/fouling-release surface. A spin-cast thin film of the b-PFMA-PEO asymmetric molecular brush exhibits a synergistic antifouling property, in which PEO side chains endow the surface with a nonfouling characteristic, whereas PFMA side chains display the fouling-release functionality because of their low surface energy. Compared with the bare surface, the protein adsorption of the surface of the coating containing asymmetric molecular brushes is reduced by 45–75%, and the cell adhesion is reduced by 70–90%, showing considerable antifouling performance.

#### 3.2.2. Silicone

Due to the high price and difficulty in preparation, fluoropolymers currently have few commercial products, and silicone polymer antifouling coatings have become a research and development hotspot. The silicone polymer has good desorption ability, and can easily make marine fouling organisms fall off by brushing or flowing. For example, PDMS has a combination of low surface energy and low elastic modulus, and it has gradually become the base of most antifouling coatings. Selim et al. [54] prepared a super-hydrophobic PDMS-Ag@SiO_2_ core-shell nanocomposite antifouling coating using the modified Stöber methods. The formation process of the coating is shown in Figure 8. The Ag@SiO_2_ core-shell nanofiller was inserted into the surface of the PDMS material by the solution casting method, and a strong coating was formed according to the hydrazination curing mechanism. The water contact angle (WCA) of the coating was determined to be 156°, and the surface free energy was 11.15 mJ/m^2^. Biological tests prove that the coating has significant inhibitory effects on different bacterial strains, yeasts and fungi.

Low surface energy coatings do not require the use of antifoulant and they generally have a smooth surface, which is of great significance for reducing the navigation resistance of the vessels and reducing fuel consumption. The obvious disadvantage of low surface energy coatings is the poor adhesion on the hull surface. In practical applications, intermediate coatings must be used to enhance the adhesion of the coatings, which complicates the coating application process and increases the cost of use. Compared with other types of coatings, low surface energy coatings are more easily destroyed. In addition, most low surface energy coatings have poor desorption ability to strong adherent organisms such as diatoms.

### 3.3. Biomimetic Antifouling Coating

Biomimetic antifouling coating, also known as microstructured surface antifouling coating, its mechanism of action is to destroy the physical attachment of marine organic matter by preparing a surface similar to the microstructure contained in the biological epidermis. For example, the skin or surface of sharks, shells, etc. [55,56] all have different structures of microstructures. This microstructured surface plays an important role in preventing or inhibiting the pollution of barnacles, algae and bacteria. The current main methods of making microstructured surfaces include laser etching [57], photolithography [58], three-dimensional (3D) printing [59], etc.

The shark skin surface structure has been extensively studied due to its drag reduction [60,61] and antifouling properties, and its various microstructures have been proven [62,63]. For example, in order to further study the antifouling effect of bionic shark skin, Pu et al. [64] prepared the biomimetic shark skin using the polydimethylsiloxane (PDMS)-embedded elastomeric stamping (PEES) method. PDMS itself is a low surface energy antifouling material, which has a certain antifouling ability, but it is prone to bioaccumulation in static or low flow state. The surface microstructure of shark skin is constructed on PDMS material, and the coating still has a low surface energy and exhibits extremely strong hydrophobicity. Air is hidden in the surface microstructure to form a thin layer of air, which can effectively reduce the interaction between the surface and protein molecules. Therefore, it is difficult for the protein to adhere to the surface and inhibit the attachment of some fouling organisms. The structure of shark skin and biomimetic shark skin is shown in Figure 9. Marine field tests show that the hydrophobicity of the PDMS material itself and the surface microstructure can effectively prevent the attachment of marine fouling organisms.

In marine environment, *Laminaria japonica* still has excellent antifouling ability in a relatively static state compared to those parade creatures. Zhao et al. [65] reported the synergistic effect between surface topography and chemical modification, and used sodium alginate and guanidine-hexamethylenediamine-PEI to replicate the surface microstructure on the surface of PDMS replicas by a layer-by-layer assembly method. The production process is shown in Figure 10. The water contact angle of PDMS with a microstructured surface is reduced to 35.3°, which is more hydrophilic. After the algae test and the bacteria test, the surface has extremely strong antifouling performance, and the antibacterial ability is up to 96.2 ± 1.3%.

Lotus leaf surface is one of the most famous superhydrophobic surfaces and displays excellent anti-biofouling performances [66,67]. Jiang et al. [68] investigate the super-repellency of the lotus leaf towards the bacterial medium, together with its mechanical bactericidal activity against the attached bacteria, and for the first time reveal the synergistic antibacterial effects of its bacterial repellency and physical rupture by nanotubes. They designed and developed a hierarchically structured superhydrophobic surface integrated with regularly spaced micro-pillar arrays and packed nanoneedles. This surface can remarkably prolong its efficiency under much harsh conditions with respect to the conventional superhydrophobic surfaces, without causing any potential antimicrobial resistance. Schematic illustration of fabrication procedure of lotus leaf-like structures is shown in Figure 11, and the microscale structure of the surface is showed in Figure 12. The surface static water contact angle exceeds 170°, and the rolling angle is less than 1°. The surface’s anti-adhesion efficiency for *E. coli* can reach more than 99%. A small amount of adhered bacteria can be completely killed, effectively overcoming the disadvantages of bacteria adhesion accumulation on the surface of a single structure sterilization. 

The biomimetic antifouling coating contains no antifoulant and will not release compounds into the sea water. It is an eco-friendly antifouling coating. However, the construction of surface microstructures is more complicated, it is difficult to repair after damage and it is less effective in real marine environments. With the rapid development of Micro-Electro-Mechanical System (MEMS) and laser repair technologies, bionic antifouling coatings are still a promising development direction.

### 3.4. Photocatalytic Antifouling Coating

The use of photocatalytic to enhance the antifouling ability of the hull has attracted the attention of many scholars [69]. When some nanocomposites are exposed to sunlight, their surfaces exhibit strong oxidizing and reducing properties. Considering both economic and ecological aspects, this method has no pollution and low cost. TiO_2_ nanocrystals are widely used materials that modify coating surfaces to provide considerable mechanical reinforcement and surface wettability. Selim et al. [70] conducted an in-depth study on photocatalytic antifouling coatings, and prepared a PDMS/TiO_2_ hybrid nanocomposite. Its preparation method and action mechanism are shown in Figure 13. When sunlight illuminates TiO_2_ nanoparticles, it will excite carriers such as electrons and holes. The photogenerated electrons are transferred from the valence band to the conduction band, while the holes remain in the valence band. The electrons in the conduction band can be reduced to superoxide anion radicals, and the holes with strong oxidizing ability can oxidize water to produce hydroxyl groups. Hydroxyl groups still have strong oxidizing properties, which can attack the unsaturated bonds of organic substances or extract H atoms of organic substances to generate new free radicals. The chain reaction is excited and the bacteria are decomposed. In addition, the modified material exhibits strong hydrophilicity, which will produce a thin water film on the surface to block the adhesion of fouling organisms.

However, because the band gap of TiO_2_ nanoparticles is 4% of sunlight, they can only absorb ultraviolet rays. Therefore, it is necessary to further improve the visible light absorption effect of the material. Zhang et al. [71] synthesized a new visible light-sensitive InVO_4_/AgVO_3_ photocatalyst with a PN node structure through ion exchange and in-situ growth process. Its mechanism of action is shown in Figure 14. When the bacteria come into contact with the catalyst, the H^+^ and O^2-^ ions catalyzed by the surface of InVO_4_/AgVO_3_ will destroy these cell walls and cell membranes, resulting in severe cell rupture, cell effluent and DNA molecule destruction. About 99.9999% of *Pseudomonas aeruginosa* (*P. aeruginosa*), *Escherichia coli* (*E. coli*) and *Staphylococcus aureus* (*S. aureus*) were killed over 0.5 InVO_4_/AgVO_3_ at 30 min.

### 3.5. Nano-Composite Antifouling Coating

Due to the diversity of the working environment and the increased use requirements, it is difficult to achieve a better antifouling effect with a single performance antifouling coating. Composite antifouling coatings have become the mainstream of current antifouling coatings research [72]. Tian et al. [73] prepared a series of composite antifouling coatings consisting of silicone elastomer and nanocomposite hydrogel. They use AgNPs as cross-linking agents to improve the compatibility of hydrogels and PDMS, combine different antifouling mechanisms together, and improve the antifouling performance of hybrid coatings. The preparation principle of the coating is shown in Figure 15. Silicone and hydrogel work together to improve the desorption ability of the coating. Nanoparticles AgNPs are not only used to enhance the compatibility of silicone and hydrogel, but also make the coating have a bactericidal effect. After soaking in seawater for two years, the surface of the coating has no barnacles nor other scaling phenomena, further verifying its antifouling ability.

Selim et al. [74] prepared a superhydrophobic coating of silicone/β-MnO_2_ nanorod composite for marine antifouling. They studied how the self-cleaning and antifouling features were affected by controlling the β-MnO_2_ nanorod preparation and distribution in the silicone matrix. The preparation method of β-MnO_2_ and the construction method of the coating are shown in Figure 16. PDMS, as the substrate of the coating, is itself a low surface energy coating. By using β-MnO_2_ nanorods to construct a rough structure on the surface of PDMS, the coating is made superhydrophobic. The composite coating prevents fouling organisms from adhering through super-hydrophobicity and low surface energy. In addition, β-MnO_2_ nanomaterials also enhance the stability of the coating to temperature and PH.

### 3.6. Other Antifouling Coatings

In addition to the several coatings introduced above, some novel research results have emerged in recent years. These novel coatings have novel ideas and are expected to become new hot spots in antifouling coatings.

#### 3.6.1. Microcapsules Coating

Li et al. [75] prepared a form of environment-friendly microcapsules through mini-emulsion polymerization as show in Figure 17. They used zinc acrylate resin to wrap the synthetic microcapsules into a coating, and investigated the slow release efficiency and antifouling effect of the coating. The microcapsules had a poly(urea-formaldehyde) (PUF) shell and a mixed core of silicone oil and capsaicin. Dimethyl silicone oil has low surface tension, good water repellency, good lubricity and has the characteristics of pollution resistance, antifouling, anti-adhesion and so on. Capsaicin extracted from natural peppers does not destroy the biological chain, and can be used as a marine antifouling agent to kill plant spores and animal larvae attached to the outer surface of ships. It has broad application prospects in the field of antifouling materials. PUF microcapsules can combine multiple ingredients to delay the release rate of silicone oil and capsaicin, and extend the service life of the coating. Since the surface of the microcapsule antifouling paint has a micro-nano raised structure and the internal material can be slowly released, it has good hydrophobicity and antifouling properties. The release rate of the antifouling agents from the coating is determined by their slow release from the microcapsules to the surrounding coating matrix, and the encapsulated biocide is protected from degradation. This research has certain guiding significance for the development of long-term antifouling coatings.

#### 3.6.2. Dynamic Surface Antifouling

Xie et al. [76] first proposed the concept of Dynamic Surface Antifouling (DSA). The dynamic surface refers to a changing surface that continuously renews itself in seawater and thus decreases the adhesion of biofouling. Based on this strategy, they developed a series of biodegradable and reproducible polymer dynamic surface coatings, which have excellent antifouling properties and mechanical properties. The dynamic surface can shorten the contact time between the organisms and the surface, and make the interaction between them close to non-adhesion, thereby reducing the interaction force between the surface and the fouling organisms, making the fouling organisms difficult to attach. For degradable polymers, the surface renewal is a spontaneous process where water flow is not necessary. They developed degradable self-polishing copolymers (DSPC), made of poly (ester-co-silyl methacrylate) [77]. They inserted ester bonds into the backbone of a silyl acrylate copolymer for the first time by copolymerizing 2-methylene-1, 3-dioxepane (MDO), tributylsilyl methacrylate (TBSM) and methyl methacrylate (MMA) via radical ring opening polymerization (RROP). The degradable main chains can significantly improve the erosion of the TBSM based copolymer and avoid swelling. Consequently, the coating has excellent antifouling properties during static immersion in seawater. In addition, this type of coating can be used as a carrier and release control system for antifouling agents.

#### 3.6.3. Oil-Infused Polymers

Oil-infused ‘slippery’ polymer surfaces and engineered surface textures have been separately shown to reduce settlement or adhesion strength of marine biofouling organisms. Kommeren et al. [78] created liquid-infused surfaces embossed with surface structures. They used a photo-embossing process to create perfluorinated oil-infused fluorinated meth (acrylate) coatings with tuneable surface topography. Figure 18 showed the changed in coating surface morphology with the addition of oil-infused. Oil-infused polymers are made by exposing bulk polymeric materials such as fluoropolymer or PDMS to an excess of a chemically-matched oil, such as silicone or perfluorinated oils. The polymers absorb the oil, leaving a thin liquid layer on the material surface and a reservoir of oil in the bulk polymer. This allows oil to diffuse to the interface and replenish the surface liquid layer as it becomes depleted. The antifouling coating of this strategy can effectively resist bacterial adhesion under static and flowing conditions. The performance of the coating can be adjusted by controlling the surface fluctuations, thus, the coating can adapt to a variety of use environments.

#### 3.6.4. Sol-Gel Coating

Sol-gel derived functional coatings are commercially available for many practical applications and are emerging as suitable non-toxic alternatives to biocidal antifouling coatings. The sol-gel process provides the capacity to include inorganic and organic components at the nanometric scale. The surface energy of sol-gel is low and through modification can change the hydrophobicity and hydrophilicity of its surface. Richards et al. [79] studied several novel transparent sol-gel materials and determined their effectiveness as antifouling coatings for potential application to marine deployed sensors, camera lenses, solar panels or other related technologies. They developed a range of sol-gel coating with increasing water surface wettability and roughness. The surface energy of antifouling coatings was altered by modification of surface chemistry, while surface topology was roughened by the incorporation of amorphous fumed silica within the sol-gel. A future application of this work would be to incorporate sol gel coatings on the optical windows of sensors to reduce early stage fouling.

#### 3.6.5. Coating Based on a Synergistic Strategy

Xie et al. [80] designed a new type of antifouling costing with a longer period of validity based on a synergistic antifouling strategy. As shown in Figure 19, the antifouling paint is made of polyacrylates, tert-butyldimethylsilyl methacrylate (TBSM), eugenol methacrylate (EM) and poly(vinylpyrrolidone) (PVP). EM can add eugenol groups to the coating covalently, so that the coating has the function of contact inhibition. PVP can enhance the hydrophilicity of the coating to block biological adhesion. Eugenol can be gradually released into the sea water to expel marine life. Methyl methacrylate (MMA) ethyl acrylate (EA) and *n*-butyl acrylate (BA) are used in the coating to enhance the mechanical properties and adhesion of the coating, thereby improving the durability of the coating. TBSM can make the coating have a stable hydrolysis rate, so that eugenol can be released stably. Antifouling tests show that the coating can effectively prevent protein adsorption, bacterial adhesion and diatom adhesion. The actual marine environment test shows that the validity period of the coating is at least 8 months. This research has created conditions for the development of environmentally friendly, efficient and long-lasting antifouling paints.

## 4. Concluding Remarks

Every year, marine fouling organisms cause huge economic losses, and antifouling coating is currently the most direct and effective means of prevention. It is important to develop stable, durable and environmentally friendly antifouling coatings. The current antifouling coatings are difficult to cover in terms of stability, toxicity and cost of use, which seriously hinders its popularization and use. In the future, the research and use of marine antifouling coating will surely develop in the direction of being highly efficient, non-toxic, non-polluting and degradable.

Antifouling coating containing antifoulant mostly outperform coatings without antifoulant. It is imperative to develop new and highly effective antifoulant to replace cuprous oxide, which is currently widely used. Obtaining natural antifouling substances from nature, especially extracting antifouling active substances from marine organisms, is the most ideal source of antifoulants. Through in-depth research on the antifouling mechanism of natural antifouling active substances, the development of highly effective and non-polluting antifouling agents is an important direction of the development of antifoulants. In addition, releasable coatings are environmentally friendly and have potential development prospects. The fouling release coating is mostly suitable for ships with high speed and fast travel, and it is less effective for ships that are moored for a long time or traveling at low speed.

Coatings that rely on a single antifouling mechanism cannot achieve the desired antifouling effect, and future trend will need to integrate multiple methods. Composite coatings that integrate multiple antifouling mechanisms will surely become a research hotspot, especially some nano-composite and modified coatings. With the use of nanotechnology, marine antifouling technology has been greatly improved. When the material reaches the nanometer level, its performance will change qualitatively, and the surface performance of the coating or the release efficiency of the antifoulant will be significantly improved. In addition, combining antifoulant with fouling release coatings is also an important development direction.

## Figures and Tables

**Figure 1 marinedrugs-18-00371-f001:**
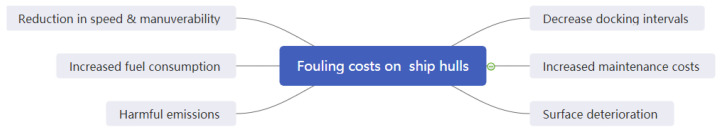
Hull attached to fouling organisms [5].

**Figure 2 marinedrugs-18-00371-f002:**
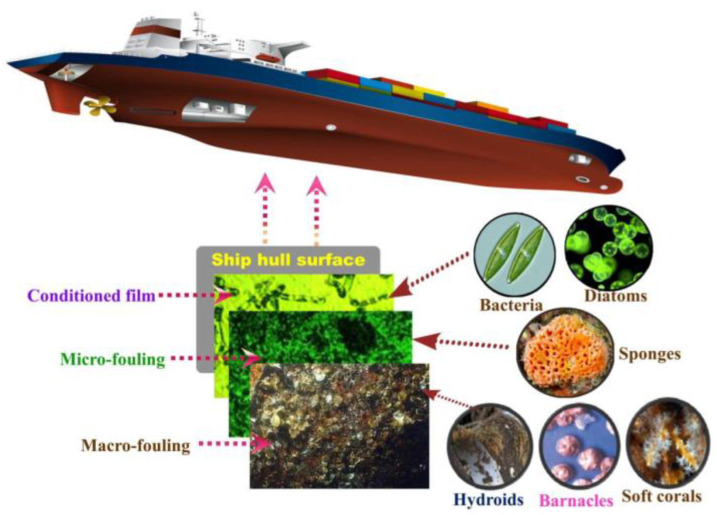
Hull surface fouling process and main fouling organisms [5].

**Figure 3 marinedrugs-18-00371-f003:**
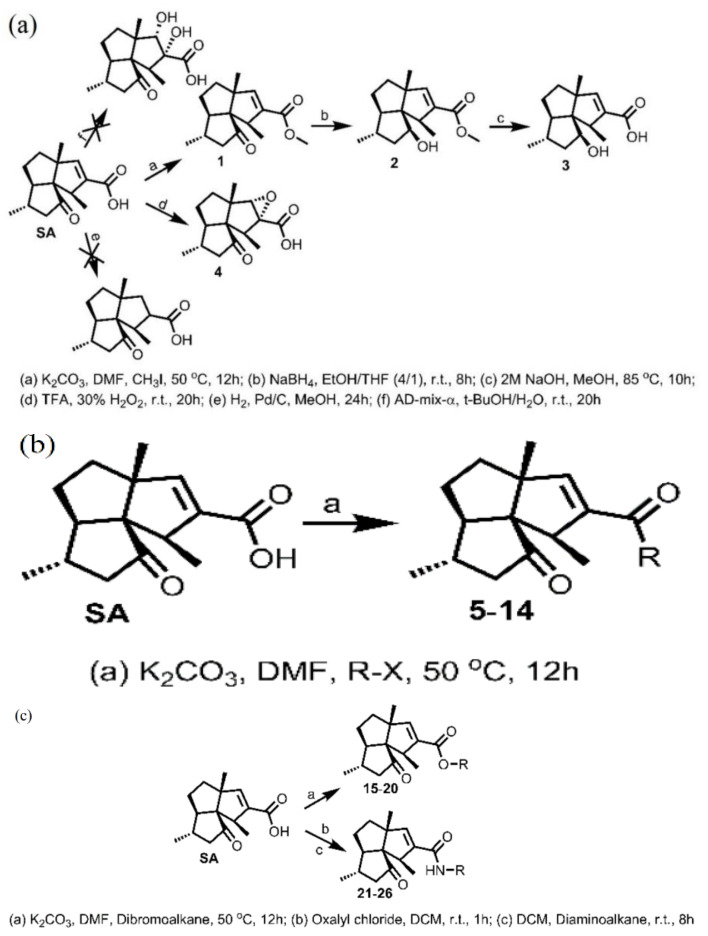
Research and preparation of subergorgic acid (SA) compounds. (**a**) Identification of bioactive functional groups of SA. (**b**) Synthesis of benzyl esters of SA. (**c**) Synthesis of SA derivatives containing methylene chain of various lengths [28].

**Figure 4 marinedrugs-18-00371-f004:**
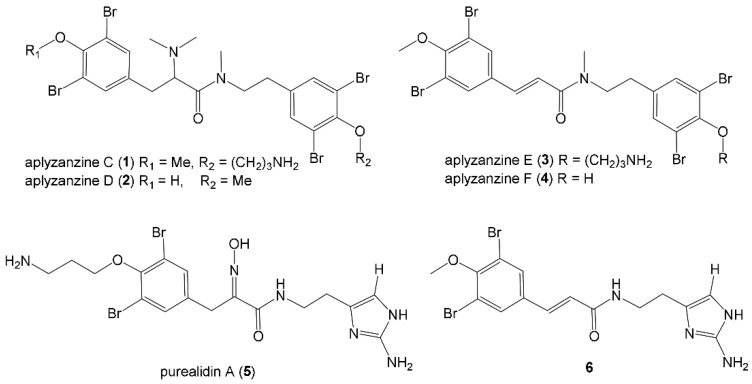
Structures of the isolated bromotyrosine metabolites [33].

**Figure 5 marinedrugs-18-00371-f005:**
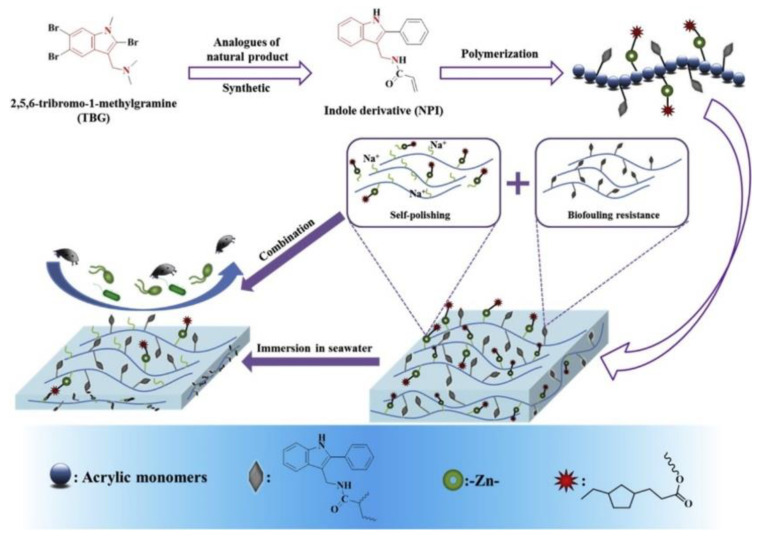
Preparation process and working mechanism of acrylate resin with indole derivatives [35].

**Figure 6 marinedrugs-18-00371-f006:**
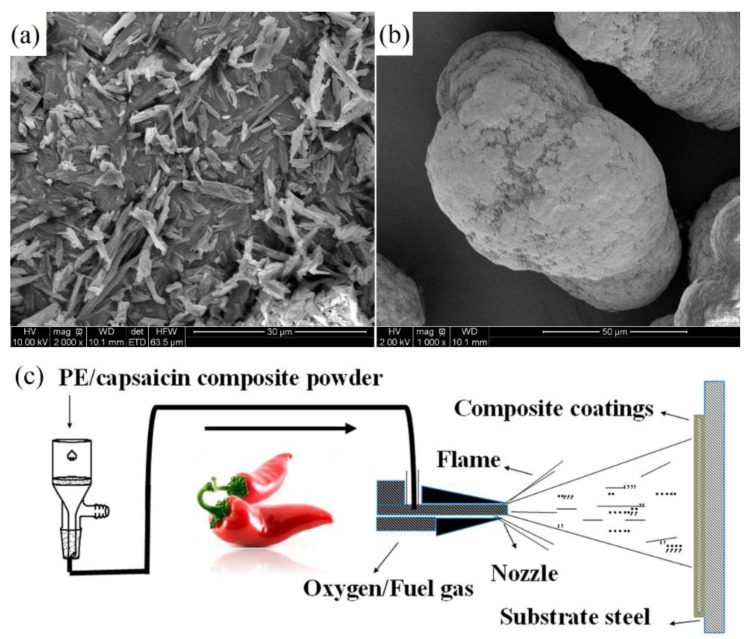
Capsaicin coating raw materials and preparation method. (**a**) FESEM image of the starting capsaicin powder; (**b**) FESEM image of the starting high-density polyethylene (HDPE) powder; (**c**) the fabrication route for the HDPE–capsaicin antifouling coatings [49].

**Figure 7 marinedrugs-18-00371-f007:**
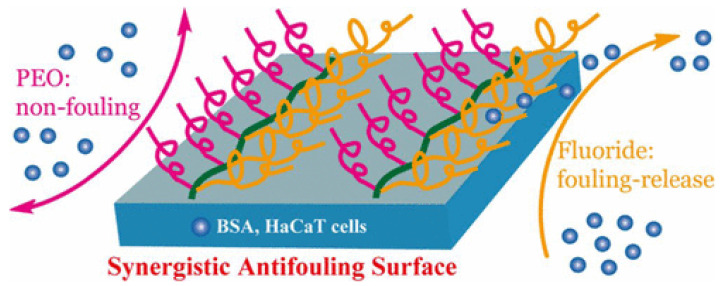
The structure of synergistic antifouling surface [53].

**Figure 8 marinedrugs-18-00371-f008:**
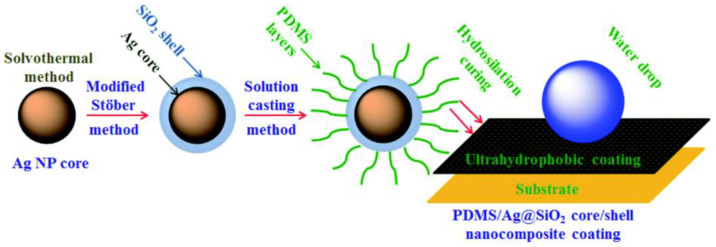
Schematic diagram of Ag@SiO_2_ core-shell nanosphere structure [54].

**Figure 9 marinedrugs-18-00371-f009:**
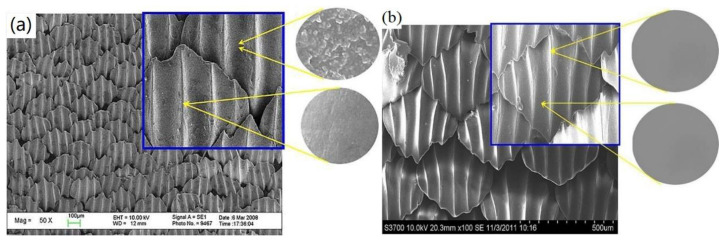
SEM images of shark skin and biomimetic shark skin: (**a**) the riblet structures of shark surface; (**b**) the surface of biomimetic shark skin prepared [64].

**Figure 10 marinedrugs-18-00371-f010:**
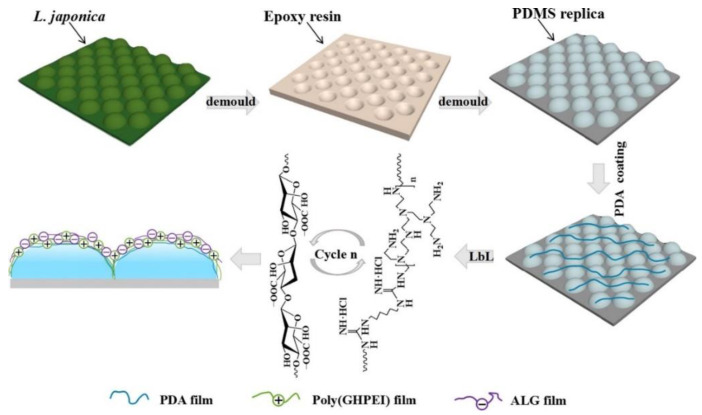
Schematic diagram of material preparation process [65].

**Figure 11 marinedrugs-18-00371-f011:**
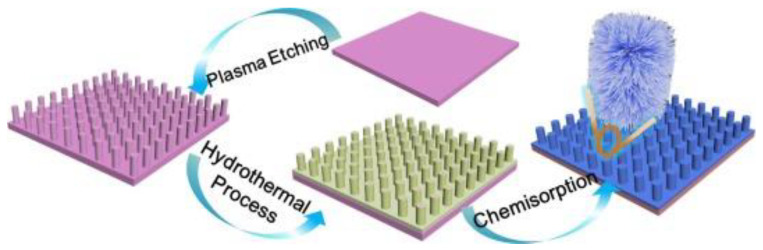
Schematic illustration of fabrication procedure of lotus leaf-like structures [68].

**Figure 12 marinedrugs-18-00371-f012:**
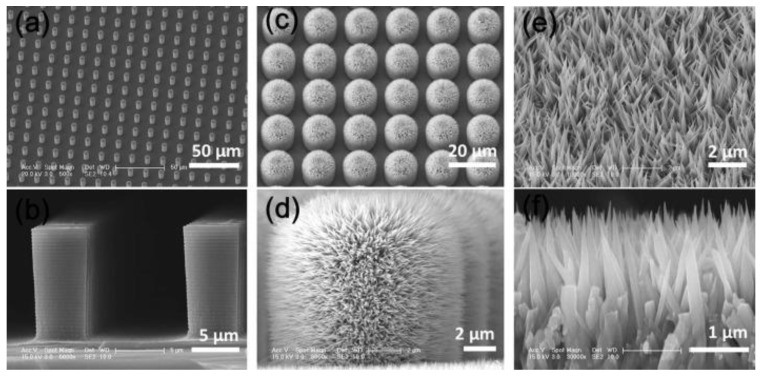
Microscale structure of different scales on the patterned surface [68].

**Figure 13 marinedrugs-18-00371-f013:**
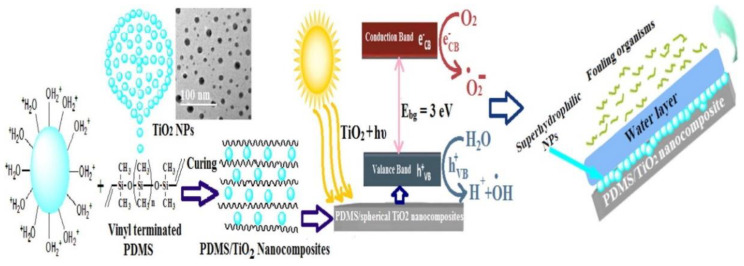
Preparation and catalytic principle of silicone/TiO_2_ nanocomposite [70].

**Figure 14 marinedrugs-18-00371-f014:**
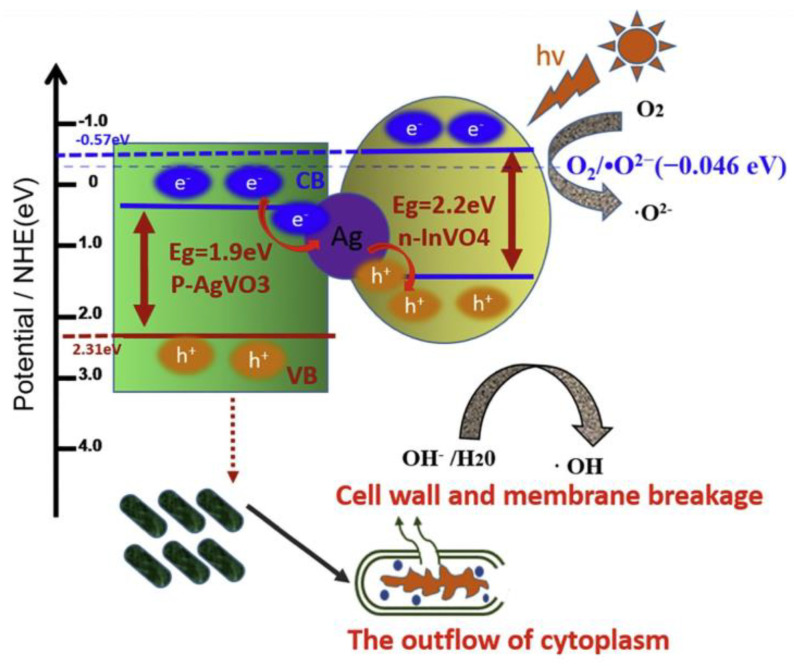
Mechanism of InVO_4_/AgVO_3_ under light [71].

**Figure 15 marinedrugs-18-00371-f015:**
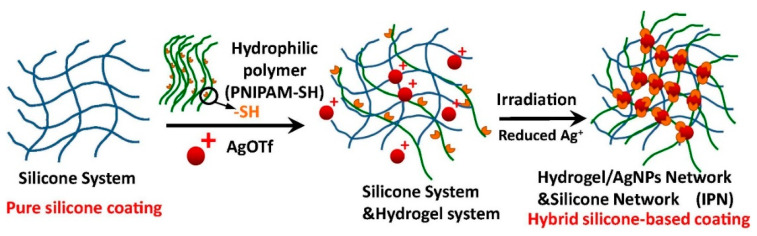
Schematic illustration of the formation of the pure silicone film and the hybrid coatings [73].

**Figure 16 marinedrugs-18-00371-f016:**
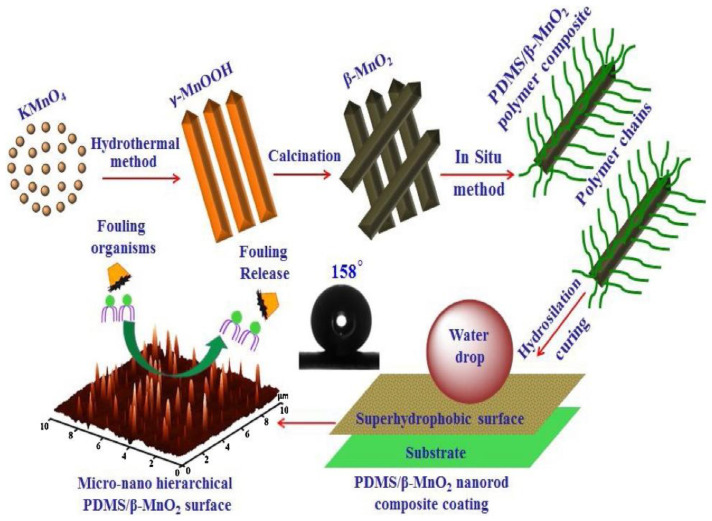
The preparation method of β-MnO_2_ and the construction method of the coating [74].

**Figure 17 marinedrugs-18-00371-f017:**
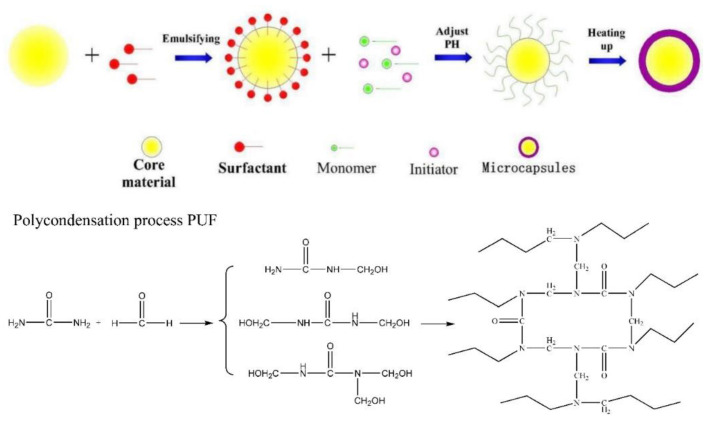
Mechanism of urea-formaldehyde microcapsule formation [75].

**Figure 18 marinedrugs-18-00371-f018:**
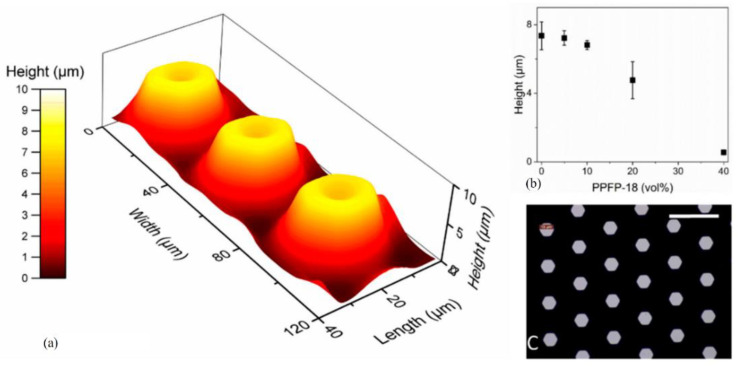
Microstructure of the coating surface. (**a**) A three-dimensional (3D) height profile of the photo-embossed features in a polymerized coating without the addition of perfluorinated oil. (**b**) Measured feature height. (**c**) Micrograph of coating surface. [78].

**Figure 19 marinedrugs-18-00371-f019:**
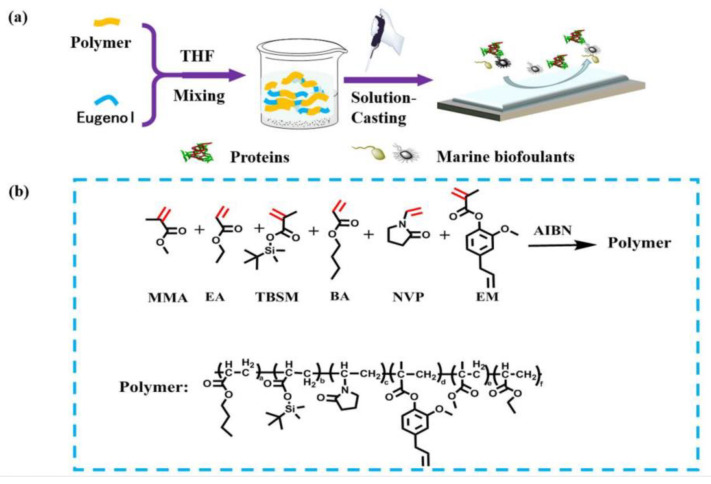
Preparation of synergistic antifouling coating [80]. (**a**) Schematic diagram of coating preparation. (**b**) Synthesis of key copolymer PAPS-EM.

**Table 1 marinedrugs-18-00371-t001:** Main research directions of marine antifouling coatings.

Type	Components	Mechanism	Characteristic	Referent
Coatings with antifoulant	Chemical antifoulant.	Poisoning or inhibiting biological growth.	Currently widely used; There are hidden environmental hazards.	[12,13,14,15,16]
	Natural product antifoulant.	Inhibiting settlement and adhesion of marine organisms.	Difficult or expensive to obtain; It is hard to retain activity.	[21,22,23,24,25,26,27,28,29,30,31,32,33,34,35,36,37,38,39,40,41,42,43,44,45,46,47,48,49]
Fouling release coatings	Silicone; Organic fluorine.	Low surface energy makes organisms difficult to attach.	Commercially available; Vulnerable; Low efficiency at static.	[50,51,52,53,54]
Biomimetic coatings	Micro-structured surfaces.	Increase the difficulty of attaching fouling organisms.	Poor effect; It is hard to be applied.	[55,56,57,58,59,60,61,62,63,64,65,66,67,68]
Other	Photocatalytic Antifouling Coating.	Photocatalysis enhances surface oxidizability and reducibility.	Have almost no effect at night and in deep sea.	[69,70,71]
	Nano-composite coating.	Strong sterilization ability; Hydrophobic.	Enhance the compatibility of other ingredients.	[72,73,74]

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
