# Peer review of "Research Strategies to Develop Environmentally Friendly Marine Antifouling Coatings"

_marinedrugs, 2020, doi:10.3390/md18070371_

Round 1

Reviewer 1 Report

This mini-review aims to highlight recent efforts that have been done mainly to develop new environmentally friendly antifouling coatings. Along the manuscript its unclear the criteria used by the authors for the selected works, namely the period of time (several papers from 2020 are missing) or if the authors intended to show only natural strategies and natural-inspired strategies that were tested in situ or strategies that were proved to be more environmental benign (features not stressed in the text). Furthermore, in section 3.1. Antifouling Coating with Antifoulant, some of the works presented do not deal with the preparation of marine coatings (see Broad comments below related to section 3.1.1 and 3.1.2). The aims should be better clarified in the abstract as well as the title should be revised to better stress the aim of the work (e.g: Research strategies to develop Environmentally Friendly Marine Antifouling Coatings? Or Natural strategies for Marine Antifouling Coatings?).

Copyrights of the figures must be acknowledge because almost all were already published before.

Some terms need uniformization: Hot spot and hotspot; anti-fouling and antifouling; anti-foulant and antifoulant.

1. Introduction

Specific comments:

- The information of each arrow in Figure 1 would be better illustrated with a scheme instead of the non-original photograph.

3. Marine antifouling coating

Broad comments:

- It is not clear if the authors want to stress commercially available coatings (some of the examples are not available in the market, others are) or if they want to show the most popular strategies that are being pursued in research.

Specific comments:

- Table 1 citation (“Table 1 summarizes the more popular environmentally friendly antifouling paints today”) and caption (Table 1. Main marine antifouling coatings) should be revised accordingly to the “broad comment above”.

- In the fourth column in Table 1, change “Night and deep sea have almost no effect” to “Have almost no effect at night and in deep sea”.

-Change “Self-polishing copolymer-based coatings containing TBT have high efficiency in preventing the settlement and growth of marine organisms so they were once the best choice.” To “Self-polishing copolymer-based coatings containing TBT had high efficiency in preventing the settlement and growth of marine organisms so they were once the best choice.”

3.1. Antifouling Coating with Antifoulant

Specific comments:

- The authors start this section in the revised Manuscript highlighting the problems of the use of cuprous oxide: “Cuprous oxide antifouling coatings have been used widely in today’s marine ship. However, it also has potential environmental risks. (…)”. However they flip to TBT as follows: “In order to prevent the antifouling coating from harming the ecological environment, the researchers tried to find useful substances from nature to replace TBT.” Here it should be changed to “(…)from nature to replace copper-based coatings”.

3.1.1. Marine Products with Antifouling Activity

Broad comments:

- From the numerous examples of marine products with antifouling activity, it is not clear why the authors selected SA and SA derivatives and tetrabromotyrosine derivatives since these compounds were not incorporated in marine coatings (this section was called: 3.1. Antifouling Coating with Antifoulant). Please check if there are some examples in the literature of marine natural products or marine natural products derivatives that have been in fact incorporated and tested in marine coatings as they would better fit this section.

Specific comments:

- The authors start this section with a redundant phrase: “Some metabolites produced by marine organisms contain certain substances that inhibit fouling biological activity, (…)”. Please change to: “Some marine organisms produce certain metabolites that inhibit fouling biological activity, (…)”.

3.1.2. Terrestrial Products with Antifouling Activity

Broad comments:

- From the selected examples, it is not clear why the authors selected chalcone derivatives from ref 40 since none of these derivatives were incorporated in marine coatings (this section was called: 3.1. Antifouling Coating with Antifoulant). It would be better placed in this section examples of terrestrial products natural products or their derivatives that have been in fact incorporated and tested in marine coatings. For example, see this paper from 2020: https://doi.org/10.1038/s41598-020-67073-8.

Despite the relevant work selected by the authors concerning terrestrial capsaicin in antifouling coatings, it seems the authors have missed another paper from 2020, concerning incorporation of capsaicin in antifouling coatings (https://doi.org/10.1016/j.scitotenv.2019.136361).

Specific comments:

Revise singular and plural: “The result shows, all of the tested compounds showed a strong inhibitory activity against barnacle settlement, with very low or moderate toxicity to non-target organisms. Its antifouling effect far exceeds that of TBT, Irgarol and copper.”

3.2. Fouling Release Coatings

Even if the authors do not intend to include advances on self-polishing marine antifouling coatings (e.g. 2020: https://doi.org/10.1021/acsami.9b22748), it could be interesting to the reader to add in the introduction of this section a comparative definition of fouling release coatings (FRC) with the previous self-polishing marine antifouling coatings (SPC), highlighting the advantages of FRC over SPC.  

3.2.1. Organic Fluorine

Specific comments:

Define PDMS, first time it appears in the text and in section 3.2.2 use only the abbreviation. Replace the abbreviation FDTS for the definition because it doesn´t appear any more in the manuscript.

3.3. Biomimetic Antifouling Coating

Specific comments:

Please change “For example, The skin or surface of sharks, shells, etc.” to “For example, the skin or surface of sharks, shells, etc.”.

3.6. Other Antifouling Coatings

3.6.2. Dynamic surface antifouling

Specific comments:

Please change “Based on this strategy, they developed a series of degradable polymer based dynamic serfaces” to “Based on this strategy, they developed a series of degradable polymer based dynamic surfaces”.

Define abbreviations DSPC, MDO, MMA via RROP, first time they appear in the text.

4. Concluding remarks

As stressed at the end of the conclusions, combining/synergistic strategies are an important development direction. However recently works of environmentally friendly marine antifouling coatings ilustrating this direction were missed in the Manuscript namely combining antifoulant with fouling release marine coating  PDMS (see for example https://doi.org/10.1016/j.ecoenv.2019.109812) and a novel and efficient antifouling coating based on a synergistic strategy, incorporating contact inhibition, fouling repelling, and antifouling properties (https://doi.org/10.1021/acs.langmuir.9b03764)

Reviewer 2 Report

This review is well prepared and well written containing the current situation of antifoulings.

Only minor editorial suggestions are as follows

  1. Page 3, line 8 from the bottom: Amphibalanus Ampitrite should be Amphibalnus amphitrite since this is a scientific name, need it to be italic
  2. Page 4, lines 8-9 from the bottom: Subergorgia suberosa should be italic
  3. Page 5, lines 1-2 from the bottom: these two lines used large capitals but they should be small capitals
  4. Page 7, line 4: Wang et al [37] should be Wang et al. [37]
  5. Page 7, line 5: Bacillus velezensis should be italic
  6. Page 8, legend of Figure 6, line 1: capsaicin coating should be Capsaicin coaging
  7. Page 8, line 4 from the bottom: coating(FRCs): add a space after coating
  8. Page 9: methacrylate(PFMA) : add a space before (PFMA)
  9. Page 11, line 4 from the bottom: AMR: may need to spell out
  10. Page 11, last line: E. coli should be italic
  11. Page 13, line 10: in-situ should be
  12. page 13, line 14: P.aeruginosaudomonas aeruginosa (P. aeruginosa) should be Paeruginosaudomonas aeruginosa
  13. Page 13, line 14: Escherichia coli (E.coli) should be E. coli
  14. Page 13, lines 14-15: Staphylococcus aureus (S.aureus) should be Staphylococcus aurenus
  15. Pages 16, 17: Figure legends: Figure should not be italic
  16. References 31, 33, 60, 69, 72, 73: titles should be small capital
  17. References 31, 33, 39, 59: scientific name should be italic
  18. Reference 61: journal name should be italic
  19. Reference 62: journal name should be appropriately abbreviated

Author Response

First of all, thank you very much for your recognition of our work, we will continue to work hard to further improve our work.

Point 1:

Page 3, line 8 from the bottom: Amphibalanus Ampitrite should be Amphibalnus amphitrite since this is a scientific name, need it to be italic

Page 4, lines 8-9 from the bottom: Subergorgia suberosa should be italic

Page 7, line 5: Bacillus velezensis should be italic

Page 11, last line: E. coli should be italic

Response 1: Thank you for pointing out this problem, we have checked our manuscript and corrected such errors. The revised content is as follows:

However, this general model cannot apply to all marine organisms. For example, the cyprids of barnacle Amphibalanus amphitrite can settle on the surface of materials without the presence of a biofilm.

Zhang et al. [27] discovered subergorgic acid (SA) from Subergorgia suberosa, which proves that it is non-toxic and has a strong inhibitory effect on attachments.

Wang et al. [37] isolated a marine proteolytic bacterial strain of Bacillus velezensis from sea mud and found that the protease produced by it had obvious inhibitory effects on barnacles

The surface's anti-adhesion efficiency for E. coli can reach more than 99%.

Point 2:

Page 5, lines 1-2 from the bottom: these two lines used large capitals but they should be small capitals

Page 8, legend of Figure 6, line 1: capsaicin coating should be Capsaicin coaging

Response 2: The revised results are as follows:

Figure 3. Research and preparation of SA compounds [28] (a) Identification of bioactive functional groups of SA. (b) Synthesis and antifouling activities of benzyl esters of SA (c) synthesis and antifouling activities of SA derivatives containing methylene chain of various lengths

Figure 6. Capsaicin coating raw materials and preparation method.

Point 3:

Page 7, line 4: Wang et al [37] should be Wang et al. [37]

page 13, line 14: P.aeruginosaudomonas aeruginosa (P. aeruginosa) should be Paeruginosaudomonas aeruginosa

Page 13, line 14: Escherichia coli (E.coli) should be E. coli

Page 13, lines 14-15: Staphylococcus aureus (S.aureus) should be Staphylococcus aurenus

Response 3: Thank you for pointing out this problem, we have corrected it.

The revised content is as follows:

Wang et al. [37] isolated a marine proteolytic bacterial strain of Bacillus velezensis from….

About 99.9999% of Pseudomonas aeruginosa (P. aeruginosa), Escherichia coli (E. coli) and Staphylococcus aureus (S. aureus) were killed over 0.5 InVO4/AgVO3 at 30 min.

Point 4:

Page 8, line 4 from the bottom: coating(FRCs): add a space after coating

Page 9: methacrylate(PFMA) : add a space before (PFMA)

Response 4: The revised content is as follows:

The fouling release coatings (FRCs) itself does not contain antifoulants

poly(2,2,2-trifluoroethyl methacrylate) (PFMA) structural units

Point 5: Page 11, line 4 from the bottom: AMR: may need to spell out

Response 5: Thank you for pointing out this problem, we have not given the definition of AMR, we have replaced AMR.

Change “ARM” to “antimicrobial resistance”

Point 6: Page 13, line 10: in-situ should be

Response 6: We checked the manuscript and reviewed the references here to confirm that in-situ is error-free and does not need to be changed.

Point 7: Pages 16, 17: Figure legends: Figure should not be italic

Response 7: We have modified it.

Point 8:

References 31, 33, 60, 69, 72, 73: titles should be small capital

References 31, 33, 39, 59: scientific name should be italic

Reference 61: journal name should be italic

Reference 62: journal name should be appropriately abbreviated

Response 8: Thank you for pointing out these issues. The revised content is as follows:

Titles should be small capital:

Borges, A.; Simoes, M. Quorum sensing inhibition by marine bacteria. Mar. Drugs 2019, 17, 427.

Saurav, K.; Borbone, N.; Burgsdorf, I.; Teta, R.; Caso, A.; Bar-Shalom, R.; Esposito, G.; Britstein, M.; Steindler, L.; Costantino, V. Identification of quorum sensing activators and inhibitors in the marine sponge sarcotragus spinosulus. Mar. Drugs. 2020, 18, 127.

Toupoint, N.; Mohit, V.; Linossier, I.; Bourgougnon, N.; Myrand, B.; Olivier, F.; Lovejoy, C.; Tremblay, R. Effect of biofilm age on settlement of mytilus edulis. Biofouling 2012, 28, 985–1001.

Tintillier, F.; Moriou, C.; Petek, S.; Fauchon, M.; Hellio, C.; Saulnier, D.; Ekins, M.; Hooper, J.N.A.; AI-Mourabit, A.; Debitus, C. Quorum sensing inhibitory and antifouling activities of new bromotyrosine metabolites from the polynesian sponge pseudoceratina n. sp. Mar. Drugs 2020, 18, 272.

Penez, N.; Culioli, G.; Perez, T.; Briand, J.F.; Thomas, O.P.; Blache, Y. Antifouling properties of simple indole and purine alkaloids from the Mediterranean gorgonian paramuricea clavata. J. Nat. Prod. 2011, 74, 2304–2308.

Scientific name should be italic:

Saurav, K.; Borbone, N.; Burgsdorf, I.; Teta, R.; Caso, A.; Bar-Shalom, R.; Esposito, G.; Britstein, M.; Steindler, L.; Costantino, V. Identification of quorum sensing activators and inhibitors in the marine sponge sarcotragus spinosulus. Mar. Drugs. 2020, 18, 127.

Tintillier, F.; Moriou, C.; Petek, S.; Fauchon, M.; Hellio, C.; Saulnier, D.; Ekins, M.; Hooper, J.N.A.; AI-Mourabit, A.; Debitus, C. Quorum sensing inhibitory and antifouling activities of new bromotyrosine metabolites from the polynesian sponge pseudoceratina n. sp. Mar. Drugs 2020, 18, 272.

Liu, H.; Chen, S.Y.; Guo, J.Y.; Su, P.; Qiu, Y.K.; Ke, C.H.; Feng, D.Q. Effective natural antifouling compounds from the plant Nerium oleander and testing. Int. Biodeterior. Biodegrad. 2018, 127, 170–177,

Zhao, L.M.; Chen, R.R.; Lou, L.G.; Jing, X.Y.; Liu, Q.; Liu, J.Y.; Yu, J.; Liu, P.L.; Wang, J. Layer-by-Layer-Assembled antifouling films with surface microtopography inspired by Laminaria japonica. Appl. Surf. Sci. 2020, 511, 145564.

Journal names have been abbreviated and revised to italic.

Round 2

Reviewer 1 Report

Considering referee’s Point 3: “The aims should be better clarified in the abstract as well as the title should be revised to better stress the aim of the work (e.g: Research strategies to develop Environmentally Friendly Marine Antifouling Coatings? Or Natural strategies for Marine Antifouling Coatings?).”

Referee: The title is now to long. Please simplify the title.

Considering author's response 10: "Natural product antifouling coatings are currently in the experimental research stage, and there are few related products. Several antifouling agents mentioned in this article are derived from natural products and have antifouling activity. When testing its antifouling effect, it was made into a simple coating, which proved that the coating made of it had good antifouling effect. "We modified the title of this section as follows: Change "3.1. Antifouling Coating with Antifoulant” to “Natural product antifoulant”."

Referee: It remains unclear the criteria used by the authors for the selection of the examples of natural product antifoulants selected from the huge number of natural product antifoulants described in recent years. It may be suggested to focus on recent examples with ecotoxicity studies included. See this example of ecotoxicity studies on marine-inspired compounds https://www.nature.com/articles/srep42424?proof=true1

Considering referee’s Point 12 “From the selected examples, it is not clear why the authors selected chalcone derivatives from ref 40 since none of these derivatives were incorporated in marine coatings” and author’s response 12: “First of all, thank you for your help in improving our thesis. The chalcone and its derivatives we introduced are a potential antifouling agent. The original author also tried to make coatings, and there may be perfect related coatings in the future.”

Referee: Once more the authors didn’t catch the work cited in ref 40 (original version): the antifouling effects of those chalcones were not tested into a simple coating. However, some ecotoxicity studies were performed in this ref and these results are relevant to be highlighted as well as every ecotoxicity study performed for each example shown in this section.

Considering author's response 20: “The two papers you presented are excellent research results, and we will add them to the citation. Among them, the synergistic antifouling coating mentioned in the second chapter is very novel, and we have organized it and added it to our manuscript (in 3.6 (5)).”

Referee: Please revise or remove the citation number 50 (revised manuscript). The place this reference was placed in your article it doesn’t make sense. In opposite, the other reference concerning synergistic antifouling coating is very well placed.

Author Response

Reply to Reviewer 1:

Thank you for asking many very meaningful questions. Some of the literature you recommended helped us a lot. According to the reviewer's opinions, we have made corresponding modifications, and we have finished the above corrections which have been marked in red in our revised manuscript as follows. The main revisions are listed as follows:

Point 1:

Considering referee’s Point 3: “The aims should be better clarified in the abstract as well as the title should be revised to better stress the aim of the work (e.g: Research strategies to develop Environmentally Friendly Marine Antifouling Coatings? Or Natural strategies for Marine Antifouling Coatings?).”

Referee: The title is now to long. Please simplify the title.

Response 1: Thanks for your suggestion. We decided to use the first title you recommended, which is highly consistent with the content of this article.

The title is as follows:

Research strategies to develop Environmentally Friendly Marine Antifouling Coatings

We have also revised the abstract as follows:

Abstract: There are a large number of fouling organisms in the ocean, which are easily attached to the surface of ships, oil platforms and breeding facilities, corrode the surface of equipment, accelerate the aging of equipment, affect the stability and safety of marine facilities and cause serious economic losses. Antifouling coating is an effective method to prevent marine biological fouling. Traditional organic tin and copper oxide coatings are toxic and will contaminate seawater, destroy marine ecology and have been banned or restricted. Environmentally friendly antifouling coatings have become a research hotspot. Among them, the use of natural biological products with antifouling activity as antifouling agents is an important research direction. In addition, some fouling release coatings without antifoulants, biomimetic coatings, photocatalytic coatings and other novel antifouling coatings have also developed rapidly. On the basis of revealing the mechanism of marine biofouling, this paper reviews the latest research strategies to develop environmentally friendly marine antifouling coatings. The composition, antifouling characteristics, antifouling mechanism and effects of various coatings were analyzed emphatically. Finally, the development prospects and future development directions of marine antifouling coatings are forecasted.

Point 2:

Considering author's response 10: "Natural product antifouling coatings are currently in the experimental research stage, and there are few related products. Several antifouling agents mentioned in this article are derived from natural products and have antifouling activity. When testing its antifouling effect, it was made into a simple coating, which proved that the coating made of it had good antifouling effect. "We modified the title of this section as follows: Change "3.1. Antifouling Coating with Antifoulant” to “Natural product antifoulant”."

Referee: It remains unclear the criteria used by the authors for the selection of the examples of natural product antifoulants selected from the huge number of natural product antifoulants described in recent years. It may be suggested to focus on recent examples with ecotoxicity studies included. See this example of ecotoxicity studies on marine-inspired compounds https://www.nature.com/articles/srep42424?proof=true1

Response 2:

The examples we have chosen basically satisfy the following conditions:

  1. Derived from or related to natural products
  2. Have antifouling activity
  3. Environmental friendly

In recent years, there have been many research results on natural antifouling agents. We prefer to use the literature in the past two years, and the various antifouling agents introduced are different in terms of source, antifouling principle and characteristics.

Many natural antifouling agents are found to ecotoxicity, so they need to be rigorously tested and evaluated. We had similar expressions in the original text (“At present, the method is still in the experimental stage, and the impact of the new compounds on the marine environment is still difficult to test.” ”countries around the world have stricter audits on bioactive products, and the impact of biofouling agents on ecosystems is still worrying”), but did not use the word "ecotoxicity". We have made minor changes to some cases to emphasize their environmental friendliness and used the term “ecotoxicity”. At the end of “3.1 natural product antifoulant”, we once again emphasized the importance of studying the ecotoxicity of natural antifouling agents (“In addition, natural product antifouling agents may have ecotoxicity, that is, toxic to non-target organisms. Therefore, for the research of natural antifouling agents, we should not only pay attention to their antifouling properties, but also explore their ecotoxicity.”).

Point 3:

Considering referee’s Point 12 “From the selected examples, it is not clear why the authors selected chalcone derivatives from ref 40 since none of these derivatives were incorporated in marine coatings” and author’s response 12: “First of all, thank you for your help in improving our thesis. The chalcone and its derivatives we introduced are a potential antifouling agent. The original author also tried to make coatings, and there may be perfect related coatings in the future.”

Referee: Once more the authors didn’t catch the work cited in ref 40 (original version): the antifouling effects of those chalcones were not tested into a simple coating. However, some ecotoxicity studies were performed in this ref and these results are relevant to be highlighted as well as every ecotoxicity study performed for each example shown in this section.

Response 3: Thanks for your comment. We have found that the previous reply is inaccurate. In Reference 40, the author introduced the antifouling activity and ecotoxicity of chalcone and its derivatives, and did not make it into an antifouling coating. The content in Reference 40 laid the foundation for the study of Reference 41. In Reference 41, chalcone derivatives were incorporated in marine coatings. We emphasize the study on the ecotoxicity of chalcone in Reference 40. The revised content is as follows:

Almeida et al. [40] studied the synthesis methods of 16 kinds of chalcone derivatives and studied the antifouling properties and ecotoxicity of chalcone through biological experiments. The results show that chalcone can effectively prevent the settlement of mussel larvae and inhibit the accumulation of other fouling microorganisms. In addition, they proved that these compounds have low ecotoxicity and clarified their great potential in the field of marine antifouling.

Point 4:

Considering author's response 20: “The two papers you presented are excellent research results, and we will add them to the citation. Among them, the synergistic antifouling coating mentioned in the second chapter is very novel, and we have organized it and added it to our manuscript (in 3.6 (5)).”

Referee: Please revise or remove the citation number 50 (revised manuscript). The place this reference was placed in your article it doesn’t make sense. In opposite, the other reference concerning synergistic antifouling coating is very well placed.

Response 4: Thanks for your comment. We have removed reference 50.

This manuscript is a resubmission of an earlier submission. The following is a list of the peer review reports and author responses from that submission.

Round 1

Reviewer 1 Report

This manuscript is a review of the status of research into antifouling properties and mechanisms. The field is overdue for a review and this is timely. However, the authors have missed an opportunity to provide this needed update. Much of the information is out of date, with only little that is truly novel presented. In addition, the authors do not show a strong working knowledge of what is available commercially and what regulations exist. For example, tributyl tin has been banned internationally since 2008 by all countries that adhere to IMO regulations. It is a centerpiece of this manuscript in discussions of present antifouling technology, when in reality it hasn't been in common use for more than a decade. Very little information is provided on modern antifouling coatings containing copper, no mention is made of newer copper free paints and the information on available commercial fouling release coatings is sparse and, in some cases, incorrect. The authors background information on the biology of fouling (section 2) is disorganized and over simplied, to the point that it begins to get facts wrong. 

I appreciate the effort that was put into finding the sources on new antifouling technology. The authors have found some newer research. However, many of the sources cited are from 5-10 years ago or more. Are these still relevant or have they been abandoned? As an example, the use of biomimetic textures. These surfaces have only ever been shown to have short term efficacy. There are well established theories on why this is having to do with the size dependence of the effect (a texture that prevents everything from bacteria all the way up to tunicates does not exist). In addition, even the organisms that use texture as a possible antifouling mechanism also use other things like surface renewal, energy or chemistries to fully prevent fouling. There are many biomimetic methods for antifouling, not just surface texture, but the authors do not mention any of the others.

Again, this a good time for a manuscript on this topic. However, this manuscript needs work before it is ready for publishing. Knowing more of the history, while still adding the novel work would be a good place to start.

Specific comments:

Line 321, you refer to aircraft, please correct to vessels or something similar.

Lines 325-326: these lines are an exact repeat. Please discard one.

Line 352: Parade creatures?

Line 417: consider starting a new paragraph for Titanium dioxide. You specifically point out that some of these photo-compounds release nothing and just become more hydrophobic. Then, with no break, you start describing TiO, which breaks down and releases oxidizing agents. This is a completely different mode of action and deserves its own paragraph.

Reviewer 2 Report

Progress of antifouling properties and antifouling mechanism were presented in this mini-review. The review was separated in two parts, the fouling mechanisms and the development on three kind of antifouling coatings

The fouling mechanisms is desribe since many years. Nevertheless it's very important to explain it in an antifouling review unlike the barnacle fouling. Moreover, it's important to remind researches which show that all step are not necessary. For example, the macrofouling may occur without a the presence of microfouling.

This last years, lot of researches have been done to develop antifouling coating in the aim to decrease the environmental impact. This review present a too small part of these researches and the papers chose are not sigificant.

To conclude, I have few remarks but the main problem in this work is the interest. At my mind the novelty on this review is to low compared to the tens reviews published since 2017 on the same topics. For a mini-review, it will more interesting to choose one kind of antifouling coatings with few any review this lats years and to develop it further, as exemple antifouling coating using natural product.